# A Simple Approach for Visual Room Rearrangement: 3D Mapping and Semantic Search

**Brandon Trabucco**
Carnegie Mellon University
`btrabucc@cs.cmu.edu`

**Gunnar A. Sigurdsson**
Amazon Alexa AI

**Robinson Piramuthu**
Amazon Alexa AI
`robinpir@amazon.com`

**Gaurav S. Sukhatme**
University of Southern California and Amazon Alexa AI
`gaurav@usc.edu`

**Ruslan Salakhutdinov**
Carnegie Mellon University
`rsalakhu@cs.cmu.edu`

## Abstract

Physically rearranging objects is an important capability for embodied agents. Visual room rearrangement evaluates an agent's ability to rearrange objects in a room to a desired goal based solely on visual input. We propose a simple yet effective method for this problem: (1) search for and map which objects need to be rearranged, and (2) rearrange each object until the task is complete. Our approach consists of an off-the-shelf semantic segmentation model, voxel-based semantic map, and semantic search policy to efficiently find objects that need to be rearranged. Our method was the winning submission to the AI2-THOR Rearrangement Challenge in the 2022 Embodied AI Workshop at CVPR 2022, and improves on current state-of-the-art end-to-end reinforcement learning-based methods that learn visual room rearrangement policies from 0.53% correct rearrangement to 16.56%, using only 2.7% as many samples from the environment.

## 1 Introduction

Physically rearranging objects is an everyday skill for humans, but remains a core challenge for embodied agents that assist humans in realistic environments. Natural environments for humans are complex and require generalization to a combinatorially large number of object configurations (Batra et al., 2020a). Generalization in complex realistic environments remains an immense practical challenge for embodied agents, and the rearrangement setting provides a rich test bed for embodied generalization in these environments. The rearrangement setting combines two challenging perception and control tasks: (1) understanding the state of a dynamic 3D environment, and (2) acting over a long horizon to reach a goal. These problems have traditionally been studied independently by the vision and reinforcement learning communities (Chaplot et al., 2021), but the advent of large models and challenging benchmarks is showing that both components are important for embodied agents.

Reinforcement learning (RL) can excel at embodied tasks, especially if a lot of experience can be leveraged (Weihs et al., 2021; Chaplot et al., 2020b; Ye et al., 2021) for training. In a simulated environment with unlimited retries, this experience is cheap to obtain, and agents can explore randomly until a good solution is discovered by the agent. This pipeline works well for tasks like point-goal navigation (Wijmans et al., 2020), but in some cases this strategy is not enough. As the difficulty of embodied learning tasks increases, the agent must generalize to an increasing number of environment configurations, and broadly scaled experience can become insufficient.

In the rearrangement setting, a perfect understanding of the environment simplifies the problem: an object is here, it should go there, and the rest can be solved with grasping and planning routines. Representing the information about the locations and states of objects in an accessible format is therefore an important contribution for the rearrangement setting. Our initial experiments suggest that accurate 3D semantic maps of the environment are one such accessible format for visual rearrangement. With accurate 3D semantic maps, our method rearranges 15.11% of objects correctly, and requires significantly less experience from the environment to do so. While end-to-end RL requires up to 75 million environment steps in Weihs et al. (2021), our method only requires 2 million

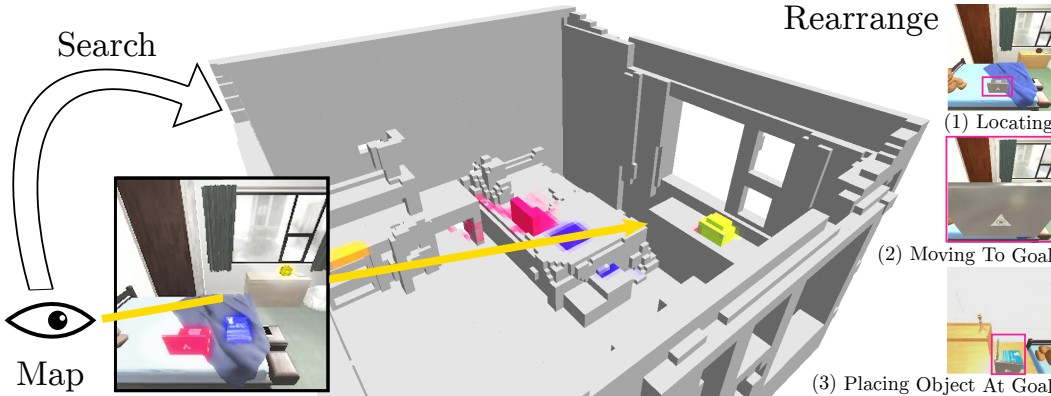

Figure 1: Our method incrementally builds voxel-based *Semantic Maps* from visual observations and efficiently finds objects using a *Semantic Search Policy*. We visualize an example rearrangement on the right with the initial position of the pink object (laptop on the bed), followed by the agent holding the object (laptop), and finally the destination position of the object (laptop on the desk).

samples and trains offline. Our results suggest end-to-end RL without an accurate representation of the scene may be missing out on a fundamental aspect of understanding of the environment.

We demonstrate how semantic maps help agents effectively understand dynamic 3D environments and perform visual rearrangement. These dynamic environments have elements that can move (like furniture), and objects with changing states (like the door of a cabinet). We present a method that builds accurate semantic maps in these dynamic environments, and reasons about what has changed. Deviating from prior work that leverages end-to-end RL, we propose a simple approach for visual rearrangement: (1) search for and map which objects need to be rearranged, and (2) procedurally rearrange objects until a desired goal configuration is reached. We evaluate our approach on the AI2-THOR Rearrangement Challenge (Weihs et al., 2021) and establish a new state-of-the-art.

We propose an architecture for visual rearrangement that builds voxel-based semantic maps of the environment and rapidly finds objects using a search-based policy. Our method shows an improvement of 14.72 absolute percentage points over current work in visual rearrangement, and is robust to the accuracy of the perception model, the budget for exploration, and the size of objects being rearranged. We conduct ablations to diagnose where the bottlenecks are for visual rearrangement, and find that accurate scene understanding is the most crucial. As an upper bound, when provided with a perfect semantic map, our method solves 38.33% of tasks, a potential for significant *out-of-the-box* gains as better perception models are developed. Our results show the importance of building effective scene representations for embodied agents in complex and dynamic visual environments.

## 2 RELATED WORK

**Embodied 3D Scene Understanding.**    Knowledge of the 3D environment is at the heart of various tasks for embodied agents, such as point navigation (Anderson et al., 2018a), image navigation (Batra et al., 2020b; Yang et al., 2019), vision language navigation (Anderson et al., 2018b; Shridhar et al., 2020), embodied question answering (Gordon et al., 2018; Das et al., 2018), and more. These tasks require an agent to reason about its 3D environment. For example, vision language navigation (Anderson et al., 2018b; Shridhar et al., 2020) requires grounding language in an environment goal, and reasoning about where to navigate and what to modify in the environment to reach that goal. Reasoning about the 3D environment is especially important for the rearrangement setting, and has a rich interdisciplinary history in the robotics, vision, and reinforcement learning communities.

**Visual Room Rearrangement.**    Rearrangement has long been one of the fundamental tasks in robotics research (Ben-Shahar & Rivlin, 1996; Stilman et al., 2007; King et al., 2016; Krontiris & Bekris, 2016; Yuan et al., 2018; Correll et al., 2018; Labbé et al., 2020). Typically, these methods address the challenge in the context of the state of the objects being fully observed (Cosgun et al., 2011; King et al., 2016), which allows for efficient and accurate planning-based solutions. In contrast, there has been recent interest in visual rearrangement (Batra et al., 2020a; Weihs et al., 2021; Qureshi et al., 2021; Goyal et al., 2022; Gadre et al., 2022) where the states of objects and the rearrangement

goal are not directly observed. In these cases, the agent is provided a direct visual input, and the environment is relatively complex and realistic. This latest iteration of rearrangement shares similarity with various other challenging embodied AI tasks such as embodied navigation (Anderson et al., 2018a; Batra et al., 2020b; Chaplot et al., 2020a; Shridhar et al., 2020; Francis et al., 2021; Min et al., 2021; Pashevich et al., 2021; Singh et al., 2021) and embodied question answering (Gordon et al., 2018; Das et al., 2018), which require finding objects and reasoning about their state.

**AI2-THOR Rearrangement Challenge.** Our work builds on the latest rearrangement methods and demonstrates how building accurate voxel-based semantic maps can produce significant gains. We focus on the AI2-THOR Rearrangement Challenge (Weihs et al., 2021), which uses AI2-THOR, an open-source and high-fidelity simulator used in many prior works (Gadre et al., 2022; Weihs et al., 2021; Shridhar et al., 2020; Gordon et al., 2018). Prior works on this challenge have studied a variety of approaches, including end-to-end RL in Weihs et al. (2021), and a planning-based approach in Gadre et al. (2022). Our approach is the first to use voxel-based semantic maps to infer what to rearrange from an experience goal as described by Batra et al. (2020a). Though both Gadre et al. (2022) and our method use planning, Gadre et al. (2022) use a graph-based continuous scene representation, and we use voxel-based semantic maps instead, which we show is more effective.

**3D Mapping & Search.** Agents that interact with an embodied world through navigation and manipulation must keep track of the world (mapping) (Thrun, 2002) and themselves (localization) (Thrun et al., 2001)—both extensively studied in robotics by processing low-level information (Engel et al., 2014), building semantic maps (Kuipers & Byun, 1991) and more recently, via techniques specifically developed to handle dynamic and general aspects of the environment (Rünz & Agapito, 2017; Rosinol et al., 2021; Wong et al., 2021). When semantics are more important than precision, such as for embodied learning tasks, recent methods have looked at neural network-based maps (Gupta et al., 2017; Chen et al., 2019; Wu et al., 2019a; Chaplot et al., 2020b; Blukis et al., 2021; Chaplot et al., 2021). Our method builds on these and adopts the use of a voxel-based semantic map and pretrained semantic segmentation model—a similar methodological setup to Chaplot et al. (2021); Min et al. (2021); Blukis et al. (2021). However, our method diverges from these prior works by using multiple voxel-based semantic maps to infer what to rearrange from an experience goal as described by Batra et al. (2020a). These prior works have instead considered geometric goals in Chaplot et al. (2021) and language goals in Min et al. (2021); Blukis et al. (2021), and ours is the first to consider an experience goal (Batra et al., 2020a). Furthermore, while a search-based policy is used in Min et al. (2021), we are the first to use search with an unspecified destination (the target object is not known a priori).

## 3 METHODOLOGY

In this section, we present a simple approach for solving visual rearrangement problems. We begin the section by discussing the visual rearrangement problem statement and metrics we use for evaluation. We then discuss our methodological contributions. First, we propose to build multiple voxel-based semantic maps representing the environment in different configurations. Second, we propose a policy that efficiently finds objects that need to be rearranged. Third, we propose a method for inferring the rearrangement goal from two semantic maps to efficiently solve visual rearrangement tasks.

**Visual rearrangement definition and evaluation metrics.** Consider the rearrangement setting defined by Batra et al. (2020a), which is a special case of a Markov Decision Process (MDP) augmented with a goal specification $g = \phi(s_0, S^*)$. This goal specification encodes the set of states $S^*$ for which the rearrangement task is considered solved from initial state $s_0$. The agent typically does not directly observe the set of goal states $S^*$, and this is reflected by the goal specification function $\phi : S \times 2^S \longrightarrow \mathcal{G}$. We consider a setting where the rearrangement goal $g$ is specified visually and the agent initially observes the environment in its goal configuration. This setting is especially challenging because the agent must remember what the environment initially looked like to infer the set of goal states. Once the goal has been understood and rearrangement has been attempted, we evaluate agents using metrics introduced by Weihs et al. (2021). We consider a *Success* metric that measures the proportion of tasks for which the agent has correctly rearranged all objects and misplaced none during rearrangement. This metric is strict in the sense that an agent receives a success of 0.0 if at least one object is misplaced—even if all others are correctly rearranged. We consider an additional *%Fixed Strict* metric that measures the proportion of objects per task correctly

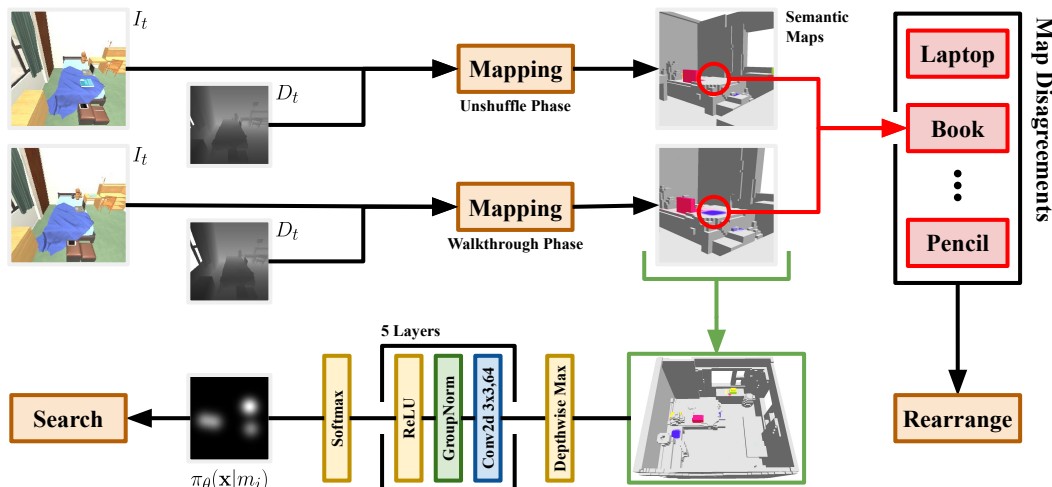

Figure 2: Our method builds voxel-based *Semantic Maps* from visual observations. Our *Semantic Search Policy* helps build accurate maps by selecting navigation goals to efficiently find objects that need to be rearranged. Once accurate maps are built, our method compares the *Semantic Maps* to identify disagreements between the maps, and rearranges objects to resolve those disagreements using a deterministic rearrangement policy.

rearranged, equal to 0.0 per task if any were misplaced. This second metric is more informative regarding how close the agent was to solving each task. Effective agents will correctly rearrange all objects in the scene to their goal configurations, maximizing their *Success* and *%Fixed Strict*.

**Building two semantic maps.** Our approach builds off recent work that uses voxel-based semantic maps in embodied settings (Blukis et al., 2021; Min et al., 2021; Chaplot et al., 2021). Our work differs from these in that we use multiple voxel-based semantic maps to encode both the goal state and current state of the environment. In particular, we build two semantic maps $m_0, m_1 \in \mathcal{R}^{H \times W \times D \times C}$ that represent 3D grids with $H \times W \times D$ voxels. Each voxel is represented with a categorical distribution on $C$ classes encoding which class is likely to occupy each voxel. Empty voxels are assigned the zero vector. In an initial observation phase, our agent navigates the scene and builds $m_0$, a semantic map encoding the goal configurations for objects in the scene. Likewise, in a second interaction phase, our agent navigates the scene and builds $m_1$, a semantic map encoding the current state of objects in the scene. At every timestep during each phase, pose, RGB, and depth images are observed, and either $m_0$ or $m_1$ is updated depending on which phase the agent is currently observing.

**Incorporating semantic predictions in the maps.** Each semantic map is initialized to all zeros and, at every timestep $t$, semantic predictions from Mask R-CNN (He et al., 2017) are added to the map. Given the RGB image observation $I_t$, we generate semantic predictions from Mask R-CNN consisting of the probability of each pixel belonging to a particular class. We filter these predictions to remove those with a detection confidence lower than 0.9 and conduct an ablation in Section 4.3. We follow Blukis et al. (2021); Min et al. (2021); Chaplot et al. (2021) and generate an egocentric point cloud $c_t^{ego}$ using the depth observation $D_t$. Each point in this point cloud is associated with a pixel in the image $I_t$ and a vector of class probabilities from Mask R-CNN. Given the current pose $x_t$, we then transform the egocentric point cloud $c_t^{ego}$ from the agent's coordinate system to world coordinate system. This transformation results in a geocentric point cloud $c_t^{geo}$ that is converted to a geocentric voxel representation $v_t^{geo} \in \mathcal{R}^{H \times W \times D \times C}$ of the same cardinality as the maps. We generate a voxelized mask $v_t^{mask} \in \mathcal{R}^{H \times W \times D \times 1}$ that equals one for every occupied voxel in $v_t^{geo}$ and zero otherwise. New semantic predictions are added to the maps with a moving average.

$$m_i[t+1] = m_i[t] \odot (1 - v_t^{mask}(1 - \epsilon)) + v_t^{geo}(1 - \epsilon) \tag{1}$$

The update in Equation 1 allows voxels to be updated at different rates depending on how frequently they are observed. The hyperparameter $\epsilon \in (0, 1)$ controls how quickly the semantic maps are updated to account for new semantic predictions, and is set to 0.5 in our experiments. An overview of how our two semantic maps are built is shown in Figure 2. We've detailed how the semantic maps are constructed from observations, and we will next describe how navigation goals are selected.

---

**Algorithm 1** 3D Mapping and Semantic Search For Visual Rearrangement

---

**Require:** visual rearrangement environment $e$, initial voxel-based semantic maps $m_0, m_1 \in \mathcal{R}^{H \times W \times D \times C}$, search-based policy $\pi_\theta(\mathbf{x}|m)$, pre-trained semantic segmentation model $g$
  **for** each phase $i \in \{0, 1\}$ **do**
    **for** each $I_t, D_t, x_t$ observed **do**
      $v_t^{geo}, v_t^{mask} \leftarrow \text{project}(\, g(I_t), D_t, x_t\,)$                           ▷ project to voxels
      $m_i[t] \leftarrow m_i[t-1] \odot (1 - v_t^{mask}(1-\epsilon)) + v_t^{geo}(1-\epsilon)$         ▷ update map
      **if** goal is reached or goal does not exist **then**
        goal $\sim \pi_\theta(\mathbf{x}|m_i[t])$                    ▷ emit a semantic search goal
      **end if**
      navigate to goal
    **end for**
  **end for**
  **while** a disagreement $d$ between $m_0$ and $m_1$ is detected **do**
    navigate to $d$ in $m_1$ and rearrange $d$ to match $m_0$
  **end while**

---

**Locating objects with a search-based policy.** Building accurate maps requires locating and observing every object in the scene so they can be added to the maps. This requires intelligently selecting navigation goals based on where objects are likely to be. We learn a high-level policy $\pi_\theta(\mathbf{x}|m_i)$ that builds off recent work in Min et al. (2021); Chaplot et al. (2021) and parameterizes a distribution over 3D search locations in the environment. The input to the policy is a 3D semantic map $m_i$ from whichever phase is currently active. The policy is a 5-layer 2D convolutional neural network that processes a 3D semantic map $m_i$ and outputs a categorical distribution over voxels in $m_i$, corresponding to 3D search locations. The policy is trained using maximum likelihood training with an expert distribution $p^*(\mathbf{x})$ that captures the locations of the $K$ objects the agent should rearrange in the current scene. This expert distribution in Equation 2 is a Gaussian mixture model with a mode centered at the location $\mu_k$ of each object, and a variance hyperparameter $\sigma^2$ for each mode.

$$p^*(\mathbf{x}) \propto \frac{1}{K} \sum_{k=1}^{K} \mathcal{N}(\mathbf{x}; \mu_k, \sigma^2 I) \tag{2}$$

Once a policy $\pi_\theta(\mathbf{x}|m_i)$ is trained that captures a semantic prior for object locations, we use planning to reach goals sampled from the policy. We build a planar graph that represents traversable space derived from voxel occupancy in the semantic map, and use Dijkstra's algorithm (Dijkstra, 1959) to find the shortest path from the agent's current location to the goal. We filter navigation goals to ensure only feasible goals are sampled, and then allow sufficient time for each navigation goal to be reached. Once the current goal is reached, we sample another goal and call the planner again.

**Inferring the rearrangement goal from the maps.** Once two semantic maps are built, we compare them to extract differences in object locations, which we refer to as map disagreements. These disagreements represent objects that need to be rearranged by the agent. To locate disagreements, we first use OpenCV (Bradski, 2000) to label connected voxels of the same class as object instances. We consider voxels with nonzero probability of class $c$ to contain an instance of that class. Object instances are then matched between phases by taking the assignment of object instances that minimizes the difference in appearance between instances of the same class. We leverage the Hungarian algorithm (Kuhn & Yaw, 1955), and represent appearance by the average color of an object instance in the map. Once objects are matched, we label pairs separated by $> 0.05$ meters as disagreements. Given a set of map disagreements $\{(x_1, x_1^*), (x_2, x_2^*), \ldots, (x_N, x_N^*)\}$ represented by the current pose $x_i$ and goal pose $x_i^*$ for each object, we leverage a planning-based rearrangement policy to solve the task. Our rearrangement policy navigates to each object in succession and transports them to their goal location. By accurately mapping with a search-based policy, inferring the rearrangement goal, and planning towards the goal, our method in Algorithm 1 efficiently solves visual rearrangement. [1]

---

[1] Code for reproducing our method is available at: `https://github.com/brandontrabucco/mass`

Table 1: Evaluation on the 2022 AI2-THOR 2-Phase Rearrangement Challenge. Our method attains state-of-the-art performance on this challenge, outperforming prior work by $875\%$ *%Fixed Strict*. Results are averaged over 1000 rearrangement tasks in each of the 2022 validation set and 2022 test set. Higher is better. A *Success* of 100.0 indicates all objects are successfully rearranged if none are misplaced and 0.0 otherwise. The metric *%Fixed Strict* is more lenient, equal to the percent of objects that are successfully rearranged if none are newly misplaced, and 0.0 otherwise.

| | Validation | | Test | |
|---|---|---|---|---|
| **Method** | **%Fixed Strict** | **Success** | **%Fixed Strict** | **Success** |
| VRR + Map (Weihs et al., 2021) | 1.18 | 0.40 | 0.53 | 0.00 |
| CSR (Gadre et al., 2022) | 3.30 | 1.20 | 1.90 | 0.40 |
| Ours w/o Semantic Search | 15.77 | 4.30 | +795% 15.11 | +900% 3.60 |
| Ours | **17.47** | **6.30** | +871% **16.56** | +1175% **4.70** |

## 4 EXPERIMENTS

In this section we evaluate our approach and show its effectiveness. We first evaluate our approach on the AI2-THOR Rearrangement Challenge Weihs et al. (2021) and show our approach leads to an improvement of $14.72$ absolute percentage points over current work, detailed in Subsection 4.1. This benchmark tests an agent's ability to rearrange rooms to a desired object goal configuration, and is a suitable choice for measuring visual rearrangement performance. Next, we show the importance of each proposed component, and demonstrate in Subsection 4.2 our voxel-based map and search-based policy exhibit large potential gains as more performant models for perception and search are developed in the future. Finally, we show in Subsection4.3 our approach is robust to the quality of object detections and budget for exploration.

**Description of the benchmark.**   In this benchmark, the goal is to rearrange up to five objects to a desired state, defined in terms of object locations and openness. The challenge is based on the RoomR (Weihs et al., 2021) dataset that consists of a training set with 80 rooms and 4000 tasks, validation set with 20 rooms and 1000 tasks, and a test set with 20 rooms and 1000 tasks. We consider a two-phase setting where an agent observes the goal configuration of the scene during an initial *Walkthrough Phase*. The scene is then shuffled, and the agent is tasked with rearranging objects back to their goal configuration during a second *Unshuffle Phase*. This two-phase rearrangement setting is challenging because it requires the agent to remember the scene layout from the *Walkthrough Phase*, to identify the rearrangement goal. Goals are internally represented by a set of valid object poses $S^* \subset (\mathcal{R}^3 \times SO(3)) \times (\mathcal{R}^3 \times SO(3)) \cdots \times (\mathcal{R}^3 \times SO(3))$, but the agent does not observe $S^*$ directly. At every time step $t$ during either phase, the agent observes a geocentric pose $x_t$, an egocentric RGB image $I_t$, and an egocentric depth image $D_t$. The rearrangement goal is specified indirectly via observations of the scene layout during the *Walkthrough Phase*. During training, additional metadata is available such as ground-truth semantic labels, but during evaluation only the allowed observations $x_t$, $I_t$ and $D_t$ can be used. Once both the *Walkthrough Phase* and *Unshuffle Phase* are complete, we measure performance using the *%Fixed Strict* and *Success* metrics described in Section 3.

### 4.1 EFFECTIVENESS AT VISUAL REARRANGEMENT

We report performance in Table 1 and show an improvement in *%Fixed Strict* from $1.9$ to $15.11$ over the current state-of-the-art method, namely Continuous Scene Representations (CSR) (Gadre et al., 2022). These results show our method is more effective than prior work at visual rearrangement. Our success of $4.63\%$ on the test set indicates our method solves 46 / 1000 tasks, whereas the best existing approach, CSR, solves 4 / 1000 tasks. Furthermore, our method correctly rearranges 499 / 3004 objects in the test set, while CSR rearranges only 57 / 3004 objects in the test set.

The results in Table 1 support two conclusions. First, 3D Mapping is a helpful inductive bias. Ours is currently the only method on the challenge to leverage 3D Mapping for identifying rearrangement goals. The next best approach, CSR Gadre et al. (2022), represents the scene with a graph, where nodes encode objects, and edges encode spatial relationships. We speculate determining which objects need to be rearranged benefits from knowing their fine-grain 3D position, which our method

Table 2: Ablation of the importance of each component of our method. Our method produces significant gains as perception and search models become more accurate. Results are averaged over 1000 rearrangement tasks in each of the 2022 validation set and 2022 test set. As in Table 1, higher is better, and a *Success* of 100.0 indicates all objects are successfully rearranged if none are misplaced and 0.0 otherwise. Our results show that as perception and search models continue to improve with future research, we have an *out-of-the-box* improvement of 34.73 *Success* on the test set.

| | Validation | | Test | |
|---|---|---|---|---|
| **Method** | **%Fixed Strict** | **Success** | **%Fixed Strict** | **Success** |
| CSR + GT T | 3.80 | 1.30 | 2.10 | 0.70 |
| CSR + GT BT | 7.90 | 3.00 | 5.90 | 2.20 |
| CSR + GT MBT | 26.00 | 8.80 | 27.00 | 10.00 |
| Ours + GT Semantic Search | 21.24 | 7.60 | +942% 19.79 | +871% 6.10 |
| Ours + GT Segmentation | 66.66 | 45.60 | +1004% 59.29 | +1707% 37.55 |
| Ours + GT Both | **68.46** | **48.60** | +1008% **59.50** | +1742% **38.33** |

directly represents via semantic maps. Second, these results also suggest that our method more successfully rearranges small objects. This is important (see additional results in Subsection 4.5) because many common objects humans use are small—cutlery, plates, cups, etc.

## 4.2 COMPONENT ABLATION

The goal of this experiment is to determine the importance of each component to our method. We consider a series of ablations in Table 2 that replace different components of our method with ground truth predictions. We first consider *Ours + GT Semantic Search*, where we substitute the predictions of our search-based policy $\pi_\theta$ with the ground truth locations of objects that need to be rearranged. We also consider *Ours + GT Segmentation*, where we substitute the predictions of Mask R-CNN (He et al., 2017) with ground truth semantic segmentation labels. The final ablation in the table *Ours + GT Both* includes both substitutions at once. In addition to reporting our performance, we reference the performance of CSR (Gadre et al., 2022) in a similar set of ablations. We consider *CSR + GT T* which uses expert trajectories that observe all objects needing to be rearranged, *CSR + GT BT* which also uses ground truth object detection labels, and *CSR + GT MBT* which additionally uses ground truth object instance pairs between the *Walkthrough Phase* and the *Unshuffle Phase*. Table 2 shows our method produces a better *out-of-the-box* improvement in all metrics as the perception and search components become more accurate, suggesting both components are important.

Table 2 demonstrates our method produces significant gains when paired with accurate semantic search and accurate semantic segmentation. When using ground-truth semantic segmentation labels and ground-truth search locations, our method attains an improvement of 35.35 absolute percentage points in *Success* compared to existing work given access to the same experts. *CSR + GT BT* makes the same assumptions as our method with both components replaced with ground-truth, and is used to compute this improvement margin. When prior work is given the *additional* accommodation of ground-truth object instance pairs between the two environment phases, *CSR + GT MBT*, our method maintains an improvement of 27.55 absolute *Success* points without the accommodation. These results show our method has greater room for improvement than prior work, with a %Fixed Strict 32.50 absolute percentage points higher than current work. Our method's room for improvement with more accurate perception and search models is appealing because accurate 3D vision models are an active area of research, and our method directly benefits from innovations in these models.

## 4.3 STABILITY VERSUS PERCEPTION QUALITY

In the previous sections, we evaluated our method's effectiveness at rearrangement, and room for growth as better perception and search models are developed. This direct benefit from improvements in perception quality resulting from better models is desireable, but an effective method should also be robust when perception quality is poor. In this section, we evaluate our method's performance stability as a function of the quality of object detections. We simulate changes in object detection

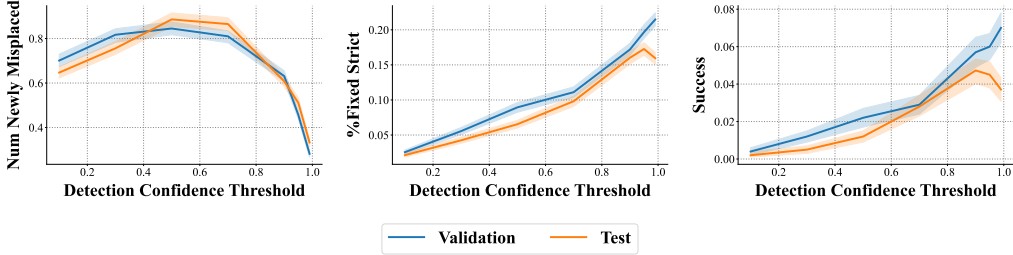

Figure 3: Rearrangement performance versus perception quality. Dark colored lines represent the average metric across 1000 tasks, and shaded regions correspond to a 68% confidence interval. Lower *Num Newly Misplaced* (left plot) is better, higher *%Fixed Strict* (center plot) and *Success* (right plot) are better. Our method improves smoothly as perception quality increases, simulated by varying the detection confidence threshold used to filter Mask R-CNN predictions detailed in Section 3.

quality by varying the detection confidence threshold of Mask R-CNN (He et al., 2017) described in Section 3. A low threshold permits accepting detections where Mask R-CNN makes high-variance predictions, reducing the quality of detections overall. In the following experiment, we vary the detection confidence threshold on the validation and test sets of the rearrangement challenge.

Figure 3 shows our method is robust to small changes in perception quality. As the detection confidence increases, simulating an improvement in object detection fidelity, performance of our method smoothly increases. Peak performance with our method on the validation set is attained with a detection confidence threshold close to 0.9, which is the value we employ throughout the paper. Error bars in this experiment are computed using a 68% confidence interval with 1000 sample points, corresponding to 1000 tasks in each of the validation and test sets. The small width of error bars indicates the observed relationship between perception quality and performance most likely holds for tasks individually (not just on average), supporting the conclusion our method is robust to small changes in perception quality. We make a final observation that as perception quality increases, fewer objects are misplaced as our method more accurately infers the rearrangement goal. These results suggest our method produces consistent gains in rearrangement as perception models improve.

## 4.4 STABILITY VERSUS EXPLORATION BUDGET

We conduct an ablation in this section to evaluate how the exploration budget affects our method. This is important because the conditions an agent faces in the real world vary, and an effective agent must be robust when the budget for exploring the scene is small. We simulate a limited exploration budget by varying the amount of navigation goals used by the agent when building the semantic maps. A lower budget results in fewer time steps spent building the semantic maps, and fewer updates to voxels described in Section 3. With fewer updates, sampling goals intelligently is crucial to ensure the agent has the information necessary to infer the task rearrangement goal.

Figure 4 shows our method is robust when the exploration budget is small. Performance is stable when less than 5 navigation goals are proposed by our semantic search module, where no penalty in *%Fixed Strict* and *Success* can be observed. This result confirms the effectiveness of semantic search: sampled goals correspond to the locations of objects likely to need rearrangement, so even when the budget is small, these objects are already observed. The experiment also shows that as the budget decreases, fewer objects are misplaced. This is intuitive because when the budget is small, fewer objects in the environment are observed and added to the map, reducing the chance of incorrect map disagreements being proposed. Additionally, when the budget is large, the agent spends the majority of the episode in navigation, and may not have enough time left to correct map disagreements, resulting in slightly lower overall performance. These results suggest our method is effective for a variety of exploration budgets, and is robust when the budget is small.

## 4.5 FAILURE MODES

Our previous experiments showed instances where our method is effective, but an understanding of its limitations is equally important. The goal of this subsection is to identify how and why our

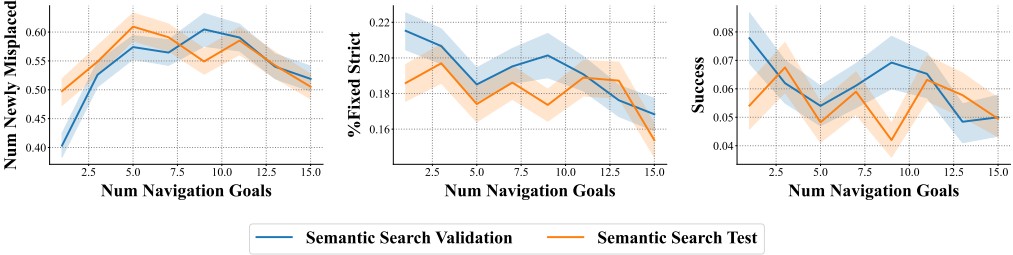

Figure 4: Rearrangement performance versus navigation budget. Dark colored lines represent the average metric across 1000 tasks, and shaded regions correspond to a 68% confidence interval. Lower *Num Newly Misplaced* (left plot) is better, higher *%Fixed Strict* (center plot) and *Success* (right plot) is better. Our method performs well even when the number of navigation goals is small and maintains a performance gain over prior work for all sizes of the budget, from one navigation goal (x-axis left) to 15 navigation goals (x-axis right).

method can fail. To accomplish this, we conduct an ablation to study how three indicators—object size, distance to the goal position, and amount of nearby clutter—affect our method. These capture different aspects of what makes rearrangement hard. For example, small objects can be ignored, objects distant to their goal can be easier to misplace, and objects too close to one another can be mis-detected. We measure the performance of our method with respect to these indicators in Figure 6 in Appendix B, and analyze the experimental conditions when our method is less effective.

Our results illuminate what kinds of tasks are difficult for our method. We find experimentally that objects further from the rearrangement goal are harder for our method to successfully rearrange. Objects within $0.326$ meters of the goal are correctly rearranged $>30\%$ of the time, whereas objects further than $4.157$ meters from the goal are only correctly rearranged $<20\%$ of the time. One explanation for this disparity in performance could be matching object instances between phases is more difficult when those instances are further apart. Better perception models can mitigate this explanation by providing more information about object appearance that may be used to accurately pair instances. While this first observation is intuitive, our second is more surprising. We find that our method rearranges small objects as effectively as large objects, suggesting our method is robust to the size of objects it rearranges. This quality is desireable because realistic environments contain objects in a variety of sizes. Effective agents should generalize to a variety of object sizes.

## 5    CONCLUSION

We presented a simple modular approach for rearranging objects to desired visual goals. Our approach leverages a voxel-based semantic map containing objects detected by a perception model, and a semantic-search policy for efficiently locating the objects to rearrange. Our approach generalizes to rearrangement goals of varying difficulties, including objects that are small in size, far from the goal, and in cluttered spaces. Furthermore, our approach is efficient, performing well even with a small exploration budget. Our experimental evaluation shows our approach improves over current work in rearrangement by 14.7 absolute percentage points, and continues to improve smoothly as better models are developed and the quality of object detections increases. Our results confirm the efficacy of active perceptual mapping for the rearrangement setting, and motivate several future directions that can expand the flexibility and generalization of the method.

One limitation of the rearrangement setting in this work is that objects only have simple states: position, orientation, and openness. Real objects are complex and have states that may change over time, potentially from interactions not involving the agent. Investigating tasks that require modelling these dynamic objects in the map is an emerging topic that can benefit from new benchmarks and methods. Another promising future direction is using an agent's experience to improve its perception. Feedback from the environment, including instructions, rewards, and transition dynamics, provides rich information about how to improve perception when true labels may be difficult to acquire. Investigating how to leverage all sources of feedback available to an agent is a useful research topic that may unlock better generalization for embodied agents in dynamic environments.

## ACKNOWLEDGEMENTS

We thank Amazon for supporting this work financially, providing access to computational resources, and feedback on intermediate drafts of this manuscript. In addition, we thank the reviewers for their suggestions and critique in the review process, which improved this paper. We thank Ben Eysenbach, Devendra Chaplot, Minji Yoon, Theophile Gervet, Murtaza Dalal, and So Yeon Min for their discussion and feedback on the paper. Finally, we thank the teams at the Allen Institute for AI that developed the AI2-THOR Rearrangement Challenge and provided benchmarking code, which has been crucial in this work.

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

APPENDIX

In this appendix we include the following supporting experiments and visualizations:

    A. We begin this appendix by presenting the performance of our map disagreement detection module for each object category. We find that our method effectively detects map disagreements for both small and large objects, and is therefore robust to object size.

    B. We then present a performance breakdown of our method for object size, distance to goal, and amount of clutter, and find that our method is less effective when objects are further from the goal or when nearby objects are closer together.

    C. We report confidence intervals for our method's performance on the rearrangement challenge.

    D. Finally, we outline the compute infrastructure needed to reproduce our experiments.

    E. We list the hyperparameters used in our paper.

    F. We categorize why our method can fail and provide a qualitative example.

The official code for our method will be released at publication.

## A   OBJECT TYPE VERSUS DETECTION ACCURACY

In this section, we visualize the relationship between the performance of our map disagreement detection module, detailed in Section 3, and the category of objects to be rearranged. For each of 1000 tasks in the validation set and test set of RoomR (Weihs et al., 2021), we record which object categories are detected as needing to be rearranged, and log the ground truth list of object categories that need to be rearranged. For each object, we calculate precision as the proportion of objects per category that were correctly identified as map disagreements out of all predicted map disagreements. Similarly, we calculate recall as the proportion of correctly identified as map disagreements out of all ground-truth map disagreements. Each bar in Figure 5 represents a $68\%$ confidence interval of precision and recall over 1000 tasks per dataset split. The experiment shows that our method is robust to the size of objects that it rearranges because small objects such as the *SoapBar*, *CellPhone*, *CreditCard*, and *DishSponge* have comparable accuracy to large objects in Figure 5.

## B   PERFORMANCE ANALYSIS

This section extends Section 4.5 with an experiment to show potential failure modes. We consider three failure modes: (1) object size, (2) object distance to the goal, and (3) closest object in the same class. These indicators are visualized in Figure 6 against *%Fixed*. Our experiment suggests our method is robust to the size of objects, shown by the lack of a global trend in the left plot in Figure 6, and confirmed by Appendix A. Additionally, the experiment shows that objects further from the rearrangement goal are solved less frequently (middle plot), which is intuitive. Instances that have been shuffled to faraway locations in the scene may require longer exploration to find, and may be more difficult for our map disagreement detection module to match. A final conclusion we can draw from this experiment is that our method can fail when object instances are too close together. This is shown in the right plot in Figure 6 by the steep drop in performance when objects in the same category are $< 1$ meter apart. In this situation, our semantic mapping module can incorrectly detect two nearby objects as a single object, which prevents their successful rearrangement. For each of these potential failure modes, better perception and mapping approaches that more accurately describe object locations and appearance can improve fidelity of our method and reduce failure.

## C   PERFORMANCE CONFIDENCE INTERVALS

We report $68\%$ confidence intervals in Table 3 to supplement our evaluation in Section 4.1 and Section 4.2. We calculate intervals using 1000 tasks from the validation and test sets of the RoomR (Weihs et al., 2021) dataset, and report the mean followed by $\pm$ interval width. Note that the official rearrangement challenge leaderboard does not expose confidence intervals, nor the sample-wise performance needed to calculate them. Due to this, we are unable to compute confidence intervals of the baselines VRR (Weihs et al., 2021) and CSR (Gadre et al., 2022) at this time. These additional results show that our improvements over prior work significantly exceed the $68\%$ confidence interval.

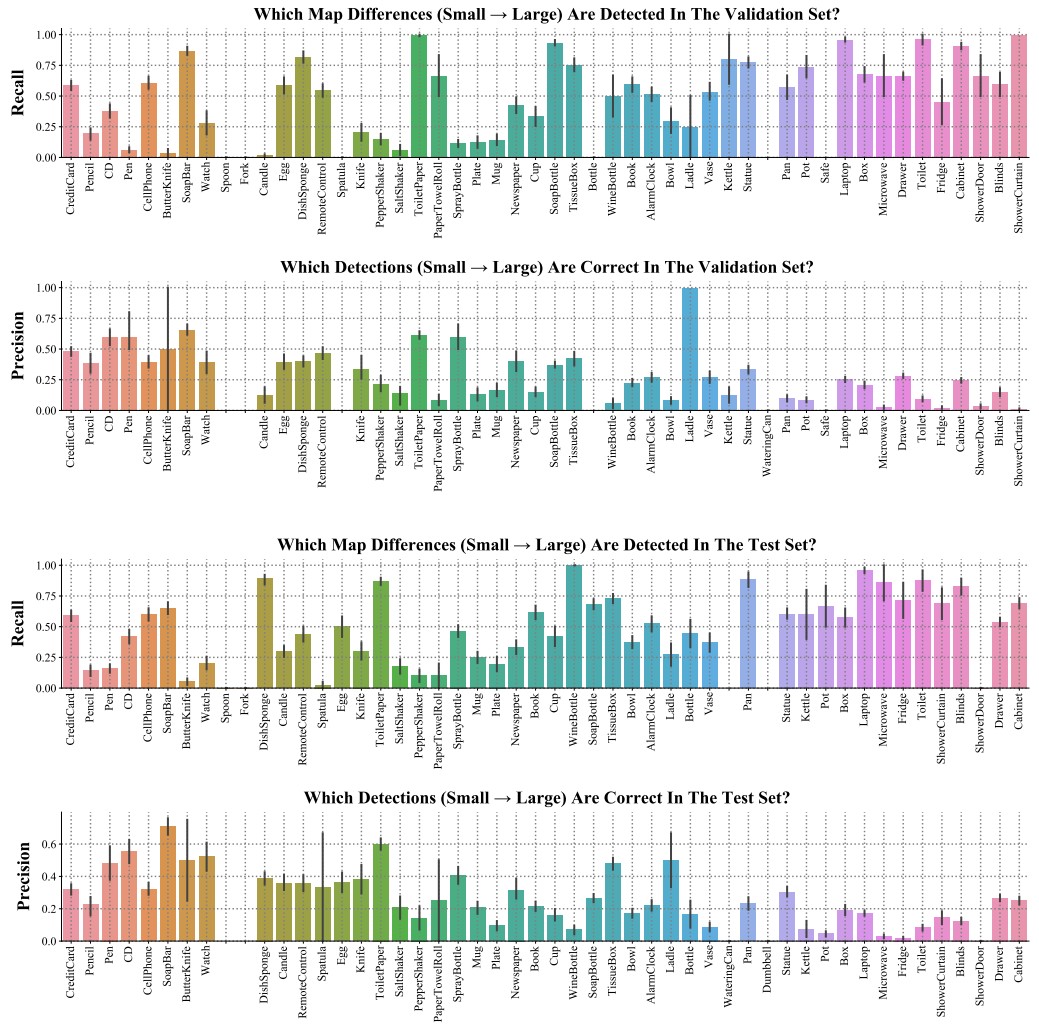

Figure 5: Performance breakdown on the validation and test sets for various types of objects. The height of bars corresponds to the sample mean of precision or recall for our map disagreement detection module. Error bars show a 68% confidence interval for each kind of object. The top two plots correspond to precision and recall on the validation set, while the bottom two plots correspond to precision and recall on the test set. Object categories are shown on the x-axis, and are ordered in ascending order of size. The experiment shows our method is robust to size, with small objects at the left end of the plots having comparable accuracy to large objects at the right end of the plots.

# D  REQUIRED COMPUTE

The goal of this section is to outline the amount of compute required to replicate our experiments. We will describe the amount of compute required for (1) training Mask R-CNN, (2) training a semantic search policy $\pi_\theta(\mathbf{x}|m_i)$, and (3) benchmarking the agent on the rearrangement challenge. For training Mask R-CNN, a dataset of 2 million images with instance segmentation labels were collected from the THOR simulator using the training split of the RoomR (Weihs et al., 2021) dataset. We then used Detectron2 (Wu et al., 2019b) with default hyperparameters to train Mask R-CNN with a ResNet50 (He et al., 2016) Feature Pyramid Network backbone (Lin et al., 2017). We trained our Mask R-CNN for five epochs using a DGX with eight Nvidia 32GB v100 GPUS for 48 hours. Our semantic search policy requires significantly less compute: completing 15 epochs on a dataset of 8000 semantic maps annotated with an expert search distribution in nine hours on a single Nvidia 12GB 3080ti GPU. Evaluating our method on the AI2-THOR rearrangement challenge requires 40

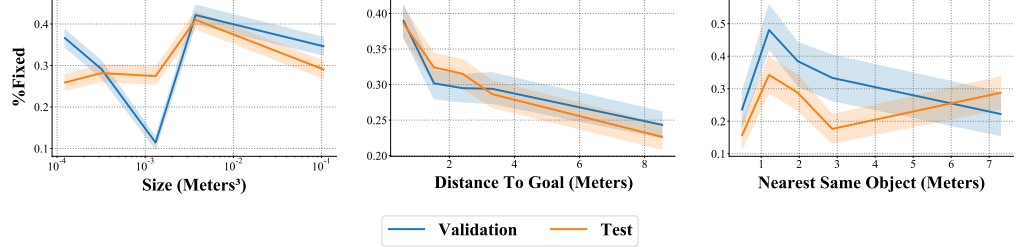

Figure 6: Performance of various ablations for different *Size (Meters³)*, *Distance To Goal (Meters)*, and *Nearest Same Object (Meters)*. These indicators measure properties of objects that make rearrangement hard. Colored lines represent the average performance over 1000 tasks in each dataset split. Error bars represent a 68% confidence interval over those same 1000 sample points. The experiment shows our method can fail when objects of the same class are too close together (right plot), and when objects are too far from the goal location, typically >4.157 meters (center plot).

Table 3: Confidence intervals for our method on the AI2-THOR rearrangement challenge. Intervals are calculated from 1000 sample points from RoomR (Weihs et al., 2021) validation and test sets. We report performance starting with the sample mean, followed by $\pm$ a 68% confidence interval width. Our improvements over prior work significantly exceed the 68% confidence interval, which suggests that our improvements are significant and our method performs consistently well.

| Method | Validation | | Test | |
|---|---|---|---|---|
| | %Fixed Strict | Success | %Fixed Strict | Success |
| Ours w/o Semantic Search | $15.77 \pm 0.85$ | $4.30 \pm 0.63$ | $15.11 \pm 0.84$ | $3.60 \pm 0.58$ |
| Ours | $17.47 \pm 0.92$ | $6.30 \pm 0.76$ | $16.56 \pm 0.89$ | $4.70 \pm 0.67$ |
| Ours + GT Semantic Search | $21.24 \pm 0.99$ | $7.60 \pm 0.83$ | $19.79 \pm 0.96$ | $6.10 \pm 0.75$ |
| Ours + GT Segmentation | $66.66 \pm 1.21$ | $45.60 \pm 1.57$ | $59.29 \pm 1.26$ | $37.55 \pm 1.53$ |
| Ours + GT Both | $68.46 \pm 1.20$ | $48.60 \pm 1.57$ | $59.50 \pm 1.31$ | $38.33 \pm 1.57$ |

GPU-hours with a 2080ti GPU or equivalent. In practice, we parallelize evaluation across 32 GPUs, which results in an evaluation time of 1.25 hours for each of the validation and test sets.

## E    HYPERPARAMETERS

We provide a list of hyperparameters are their values in Table 4. These hyperparameters are held constant throughout the paper, except in ablations that study the sensitivity of our method to them, such as Section 4.3. Our ablations show our method is robust to these hyperparameters.

## F    REASONS FOR TASK FAILURES

This section explores the reasons why certain tasks in the validation and test sets are not solved by our method. We consider four reasons for task failures that cover all possible outcomes: (1) the agent correctly predicts which objects need to be moved where, but fails to rearrange at least one object, (2) the agent incorrectly predicts an object needs to be rearranged that doesn't, (3) the agent runs out of time, and (4) the agent misses at least one object that needs to be rearranged. We visualize the proportion of failed tasks for each category in Figure 7. We find that our method with ground truth perception and search (*Ours + GT Both*) tends to fail to rearrange objects after correctly identifying which objects need to be rearranged. In contrast, the largest reason for failure for our method (*Ours*) is the agent running out of time, followed by rearranging incorrect objects. This suggests the largest potential gains for our method arise from improving the speed and fidelity of map building, whereas, the optimality of the rearrangement policy becomes the bottleneck once a perfect map is available.

Table 4: Hyperparameters used by our approach for all rearrangement tasks.

| Hyperparameter | Value |
| --- | --- |
| voxel size | 0.05 meters |
| map height $H$ | 384 |
| map width $W$ | 384 |
| map depth $D$ | 96 |
| classes $C$ | 54 |
| detection confidence threshold | 0.9 |
| rearrangement distance threshold | 0.05 meters |
| expert search distribution $\sigma$ | 0.75 meters |
| $\pi_\theta$ convolution hidden size | 64 |
| $\pi_\theta$ convolution kernel size | $3 \times 3$ |
| $\pi_\theta$ layers | 5 |
| $\pi_\theta$ activation function | ReLU |
| $\pi_\theta$ optimizer | Adam |
| $\pi_\theta$ learning rate | 0.0003 |
| $\pi_\theta$ batch size | 8 |
| $\pi_\theta$ epochs | 15 |
| $\pi_\theta$ dataset size | 8000 |

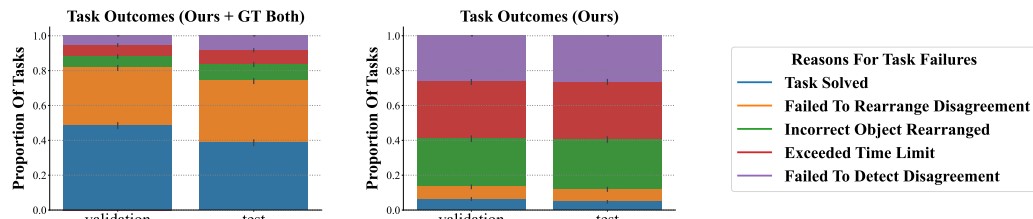

Figure 7: Categorization of the reasons why our method fails to solve tasks. The proportion of tasks that are solved (shown in blue) or fail due to one of four reasons (orange, green, red, purple) is shown for different ablations of our method. The height per bar corresponds to the proportion of tasks in the validation or test set in each category, and error bars indicate a $68\%$ confidence interval. This experiment shows the largest reason for failure is a result of mapping errors. In the right plot, the agent fails most frequently by rearranging the wrong object, and by running out of time, which can result from imperfect semantic maps. In contrast, once perfect maps are available in the left plot, the largest source of errors are due to an imperfect planning-based rearrangement policy instead.

## G  IMAGE FEATURES FOR MATCHING OBJECT INSTANCES

We conduct an experiment where we use image features for matching instances of objects between phases instead of their average color. We use ResNet50 (He et al., 2016) pretrained on ImageNet, and compute a mean image feature $h_i \in \mathcal{R}^{256}$ for each object. We process images with the ResNet50 backbone, extract a spatial feature map after the first residual block of ResNet50, and back-project the features to a voxel grid. In this fashion, we introduce 256 additional channels at each voxel to store the image features. We then compute $h_i$ by averaging the voxel features corresponding to occupied voxels for object $i$ in the map. Results in Table 5 show that matching object instances using image features improves the performance of our method by 6.97 %Fixed Strict on the test set.

## H  EFFECT OF SEMANTIC SEARCH ON FOUND OBJECTS

To understand how our Semantic Search policy leads to an improvement in downstream performance, our hypothesis is that Semantic Search leads the agent to find more objects during episodes. We

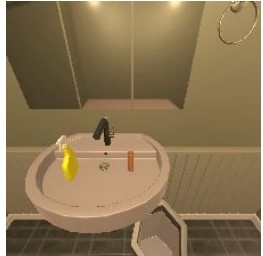 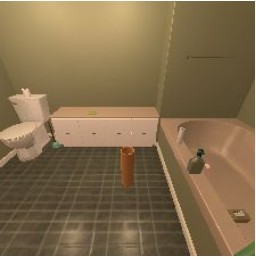 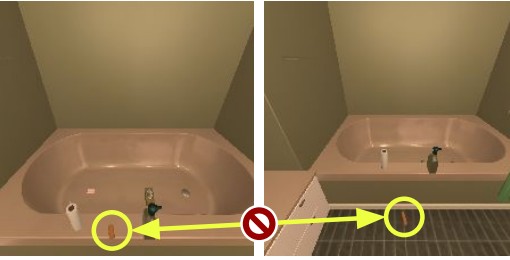

Locating *ToiletPaper*    Moving To Goal    Attempt Placing At Goal    Placed At Wrong Location

Figure 8: Qualitative example for why rearranging the correct object can fail. In this task, the agent correctly predicts the *ToiletPaper* needs to be rearranged, but fails to place the *ToiletPaper* in the correct location. The rightmost image shows the goal is located on the floor, but the agent mistakenly places the *ToiletPaper* on the bathtub instead, shown in the second image from the right.

Table 5: Matching object instances using image features. Results show that our method can be further improved by matching object instances between the walkthrough and unshuffle phases to maximize similarity of corresponding features from ResNet50 (He et al., 2016) pretrained on ImageNet.

| | Validation | | Test | |
|---|---|---|---|---|
| **Method** | **%Fixed Strict** | **Success** | **%Fixed Strict** | **Success** |
| Ours | $17.47 \pm 0.92$ | $6.30 \pm 0.76$ | $16.56 \pm 0.89$ | $4.70 \pm 0.67$ |
| Ours + Feature Matching | $23.06 \pm 1.04$ | $7.81 \pm 0.88$ | $23.59 \pm 1.03$ | $6.79 \pm 0.82$ |

test this hypothesis by measuring the percent of objects found during episodes of the official 2022 Rearrangement Challenge. We consider an object found once the agent navigates within 1 meter of the object. We track the cumulative percent of objects found, and report in Figure 9 the mean and 68% confidence interval of this metric across 1000 test set tasks. The results confirm our hypothesis: in both phases, Semantic Search leads the agent to find more objects faster. In particular, at 250 episode timesteps during the Walkthrough phase, the agent has a $9.13 \pm 1.23$ higher percent than a uniform baseline. During the Unshuffle phase, the agent has a $6.23 \pm 1.20$ higher percent than our uniform baseline at 100 timesteps. This improvement becomes less significant as time increases during the Unshuffle phase, suggesting Semantic Search is most helpful with a small time budget.

## I NOISY POSE ESTIMATION

We simulate noisy pose estimation due to imperfect localization and mapping using the sensor noise model introduced by Sarch et al. (2022), which builds on earlier work from Chaplot et al. (2020b), and constructs a Gaussian distribution based on real robot localization data collected from a real LocoBot robot (Murali et al., 2019), and applies this noise model in simulation. This model adds Gaussian noise with $\sigma = 0.005$ meters to positional observations, and adds Gaussian noise with $\sigma = 0.5$ degrees to yaw observations at each timestep. Results with noisy pose are given in Table 6. Our method is robust to this sensor noise model. On both the validation and test sets of the 2022 AI2-THOR Rearrangement Challenge, sensor noise leads to a slight improvement in performance, which we attribute to noise producing a smoother semantic map.

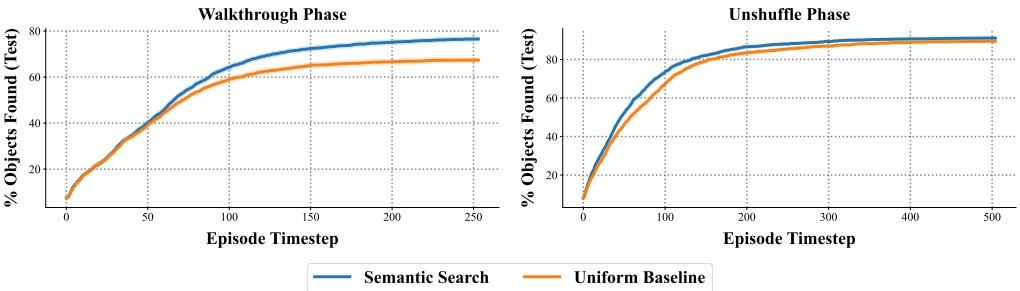

Figure 9: Impact of Semantic Search on the percent of shuffled objects found during either phase. In both cases, we observe an increase in the percent of shuffled objects found when using Semantic Search. The improvement is most significant during the walkthrough phase, where Semantic Search leads to an improvement of +9.13 percent of objects found at 250 episode timesteps. The improvement during the unshuffle phase is smaller, with +6.23 at 100 timesteps, and +1.67 at 500 timesteps.

Table 6: Impact of imperfect localization due to sensor noise on our method.

| Method | Validation | | Test | |
|---|---|---|---|---|
| | %Fixed Strict | Success | %Fixed Strict | Success |
| Ours | 17.47 | 6.30 | 16.56 | 4.70 |
| Ours + Noisy Pose | 19.84 | 6.80 | 17.33 | 4.90 |

