# OpenReview forum: "A Simple Approach for Visual Room Rearrangement: 3D Mapping and Semantic Search"
_ICLR.cc/2023/Conference — ICLR 2023 poster_

### Official Review · Reviewer_Vjug · 2022-10-20

**Confidence:** 4
**Correctness:** 3
**Technical Novelty And Significance:** 3
**Empirical Novelty And Significance:** 3
**Recommendation:** 6

**Clarity, Quality, Novelty And Reproducibility:**

Although the idea of a voxel-based semantic map is highly similar to previous works, the authors proposed to employ such an idea in the two-stage room rearrangement task and demonstrate the effectiveness of the proposed method. As a result, the proposed method is self-motivated and reasonable. Moreover, the paper is well written and easy to follow, except for some missing details and confusing parts:

-) Missing details:

(a) Is the MaskRCNN used in the proposed method fine-tuned in the environment? If so, what are the details about the training/validation/test scene splits, and how much data is used to fine-tune?

(b) Is the mask v_t^{mask} also generated by the segmentation predictions (e.g., registered as an unoccupied voxel by the background class and occupied voxels otherwise)?

(c) It is unclear how many steps are used in the “navigate to goal” operation in Algorithm 1. If the agent can use many steps to reach the goal location based on the estimated shortest path, it results in an unfair comparison to baselines. On the other hand, at the end of the algorithm (the while-loop), the agent seems allowed to take unlimited steps for a single goal. Again, it would pose an unfair comparison to the baselines.

(d) Some implementation details are missing, such as the action space used in the environment, the number of allowed steps for an episode, etc.

-) Confusing parts:

(a) The notation in Algorithm 1 is confusing. The notation “g” for the segmentation model has been used for the goal specification defined at the beginning of Section 3. The notation “D” sometimes refers to depth observation and sometimes refers to the size of the reconstructed map.

(b) In the paragraph “Inferring the rearrangement goal from the maps”, the authors mentioned considering voxels with “nonzero probability of class c” to contain an instance of that class. However, it is unclear how to determine a voxel’s class when the voxel carries many nonzero probabilities over many classes.

(c) In the last paragraph in section 4.2, the authors mentioned that the proposed method with both ground-truth segmentation and ground-truth search locations achieves a 35.35 absolute percentage improvement in Success over the existing work given by the same experts. However, is not the 35.35 improvement from “Ours+GT Segmentation - CSE + GT BT”, which does not use ground-truth search locations?

**Strength And Weaknesses:**

+) The idea of the proposed method is self-motivated and interesting. The method first reconstructs the screen into a 3d semantic map in the walkthrough stage to store the necessary information. Later, the method reconstructs the scene into another 3d semantic map in the execution stage and compares the disagreement between the two maps. In this way, the agent can not only explicitly record the misplaced object information in the scene but explore the environment and understand its traversable space.

+) The performance improvements on the challenging task over a competing baseline are significant. An ablation study in Table 2 further shows that this two-stage semantic map reconstruction idea has more room to improve over the previous graph-based approach. Moreover, various studies (e.g., different detection confidence thresholds, different numbers of navigation goals, different object sizes, different initial distances to the goal, and different nearest same object category distances) confirm the method’s effectiveness.

+) The proposed method only needs 2 million samples, thanks to offline training. It is very sampling efficient compared to the online RL training counterpart, which requires 75 million environment steps.

-) The idea of voxel-based semantic map reconstruction is highly similar to several works of literature (Min et al., 2021; Chaplot et al., 2021), although the authors point out that the previous works do not reconstruct multiple maps to determine the disagreement between them. However, previous works do not produce multiple maps because the interested tasks do not require it.

-) Based on Table 2, we can find that the proposed method highly relies on the quality of the semantic map. The fidelity of such a 3d semantic map depends on the perception model (MaskRCNN) and the agent pose (ground truth pose in this work). It is unclear whether the model can maintain effectiveness when facing unseen objects or a noisy environment (noisy pose estimation).

**Summary Of The Paper:**

This work proposes establishing high-fidelity semantic 3d maps of a scene for the visual room rearrangement task. The method generates two maps, including one for the scene at the goal configuration in the walkthrough stage and the other one for the scene at unshuffle stage. A post-processing is further used to connect the same object instances between two maps. Further, the method introduces a high-level planner, namely the search-based policy, to decide which object needs to be rearranged so that a low-level policy could be used to find the shortest path to the target object or location on the built map. The proposed method achieves state-of-the-art in the 2-phase RoomR dataset, and various ablation studies demonstrate its effectiveness.

**Summary Of The Review:**

This work demonstrates that high-fidelity semantic voxel-based maps could significantly improve the visual room arrangement task performance. The idea is self-motivated, and the results are encouraging and exciting. Moreover, the ablation study also shows that the proposed method has more room for improvement by having a better perception model. I would recommend this work to the conference with more clarification of the missing details and confusing parts mentioned above.

---

> ### Author Response · Authors · 2022-11-10
> **Response To Reviewer Vjug (1/3)**
>
> We thank Reviewer Vjug for their assessment of the paper.
>
> * “-) The idea of voxel-based semantic map reconstruction is highly similar to several works of literature (Min et al., 2021; Chaplot et al., 2021), although the authors point out that the previous works do not reconstruct multiple maps to determine the disagreement between them. However, previous works do not produce multiple maps because the interested tasks do not require it.”
>
> Our novelty lies in the problem statement: rearranging objects to a goal state, when the goal state is only passively observed during an initial walkthrough phase. Prior works that build voxel-based semantic maps, including Min et al., 2022, Blukis et al. 2021, and Chaplot et al. 2021, assume the task goal is specified extrinsically. For example, in Min et al., 2022 and Blukis et al. 2021 the task goal is specified to the agent via a prompt. Similarly, in Chaplot et al. 2021 the task goal is specified via an object category. However, in our setting, the agent must act to observe the goal, and lacking proper exploration during the walkthrough phase, the agent may not observe the goal at all. Our unique problem statement requires our method to reason about scene changes. We are the first to consider the use of voxel-based semantic maps in a setting where the agent must explore and reason about scene changes to observe the task goal.
>
> Our goal is to find the simplest method capable of serving as a strong baseline for our unique problem statement. All reviewers agree the method’s simplicity is a strength. Our use of learned policies with voxel-based semantic maps stems from our desire for simplicity. Our method does not require more sophistication to perform well in practice.
>
> * “-) Based on Table 2, we can find that the proposed method highly relies on the quality of the semantic map. The fidelity of such a 3d semantic map depends on the perception model (MaskRCNN) and the agent pose (ground truth pose in this work). It is unclear whether the model can maintain effectiveness when facing unseen objects or a noisy environment (noisy pose estimation).”
>
> We are conducting a new ablation that simulates noisy pose estimation using a realistic noise model collected from SLAM pose estimates on a real robot (Chaplot et al., 2020). We will include this ablation in the final version of the manuscript. Additionally, generalization to unseen objects is an interesting and relevant challenge, but is not considered as part of the RoomR task definition. This dataset is constructed with training/validation/testing splits such that all classes of objects that must be rearranged in the validation/testing sets are present in the training set. Visual Room Rearrangement is hard despite not testing generalization to unseen classes of objects. In follow-up work, we are considering generalization to unseen objects, but due to the nature of the RoomR task definition, we consider such an evaluation out-of-scope for this paper.
>
> * “(a) Is the MaskRCNN used in the proposed method fine-tuned in the environment? If so, what are the details about the training/validation/test scene splits, and how much data is used to fine-tune?”
>
> Correct, our MaskRCNN is trained using 2 million images with instance segmentation labels, collected from the THOR simulator on the RoomR dataset. These details, and others are described in Appendix D, Required Compute. The training/validation/test scene splits used in this work are the same splits established by Weihs et al., 2021, and can be found in the associated benchmark code for the 2022 rearrangement challenge. Additionally, on acceptance of the paper, we are releasing official source code, which replicates all experiments, including the instance segmentation dataset creation, the training of Mask R-CNN using Detectron2, and the evaluation of our method on RoomR.
>
> * “(b) Is the mask $v_t^{mask}$ also generated by the segmentation predictions (e.g., registered as an unoccupied voxel by the background class and occupied voxels otherwise)?”
>
> $v_t^{mask}$ is equal to one at every location in $v_t^{geo}$ where a voxel is occupied, and is zero otherwise. Assuming that $v_t^{geo} \in \mathcal{R}^{H \times W \times D \times C}$ is a numpy array, the following snippet $v_t^{mask}$ = np.not_equal($v_t^{geo}$, 0).any(axis=3, keepdims=True).astype(np.float32) produces the mask. Additionally, on acceptance of the paper, we are releasing official source code, which can be inspected to clarify how this mask is used during our map-building.

---

> > ### Author Response · Authors · 2022-11-10
> > **Response To Reviewer Vjug (2/3)**
> >
> > * “(c) It is unclear how many steps are used in the “navigate to goal” operation in Algorithm 1. If the agent can use many steps to reach the goal location based on the estimated shortest path, it results in an unfair comparison to baselines. On the other hand, at the end of the algorithm (the while-loop), the agent seems allowed to take unlimited steps for a single goal. Again, it would pose an unfair comparison to the baselines.”
> >
> > All methods and baselines are evaluated according to the official 2022 Rearrangement Challenge task specification, which allows for a maximum of 250 environment transitions during the first phase, and 500 during the second phase (750 total). If our agent takes too many steps during the “navigate to goal” operation in Algorithm 1, the episode is forcibly terminated, and the next phase / task begins. Baselines are evaluated with the same termination condition, ensuring the evaluation is fair for all. Algorithm 1 will be updated in the manuscript to reflect this clarification to the method. Additionally, on acceptance of the paper, we are releasing official source code, which clarifies how many steps are usually taken during the “navigate to goal” operation.
> >
> > * “(d) Some implementation details are missing, such as the action space used in the environment, the number of allowed steps for an episode, etc.”
> >
> > All methods and baselines are evaluated according to the official 2022 Rearrangement Challenge task specification, which allows for a maximum of 250 environment transitions during the first phase, and 500 during the second phase (750 total). Additionally, the action space for the agent is defined by the official 2022 Rearrangement Challenge, and can be found in the publicly available code for the official benchmark. Summarizing it here, the action space consists of four move actions in the cardinal directions, two rotation actions that increment yaw by 90 degrees, two standup and crouch actions that increment height by 0.675 meters, two look-up and look-down actions that increment camera elevation by 30 degrees each, a set of pickup actions corresponding to each interactable object class, and a drop action.
> >
> > * “(a) The notation in Algorithm 1 is confusing. The notation “g” for the segmentation model has been used for the goal specification defined at the beginning of Section 3. The notation “D” sometimes refers to depth observation and sometimes refers to the size of the reconstructed map.”
> >
> > We thank the reviewer for noticing this and will use distinct letters in the final manuscript.
> >
> > * “(b) In the paragraph “Inferring the rearrangement goal from the maps”, the authors mentioned considering voxels with “nonzero probability of class c” to contain an instance of that class. However, it is unclear how to determine a voxel’s class when the voxel carries many nonzero probabilities over many classes.”
> >
> > When processing the maps to extract a set of map disagreements, we allow voxels to contain multiple classes. Voxels are considered to contain class $c$ if the vector of class probabilities present at that voxel has >0 probability of containing class c. Situations where voxels have > 0 probability of containing multiple classes can arise when Mask R-CNN detects different objects at the same map location from different views. Details for this processing step will be available in our official code release on acceptance of the paper, which exactly reproduces our Map Disagreement calculation.
> >
> > * “(c) In the last paragraph in section 4.2, the authors mentioned that the proposed method with both ground-truth segmentation and ground-truth search locations achieves a 35.35 absolute percentage improvement in Success over the existing work given by the same experts. However, is not the 35.35 improvement from “Ours+GT Segmentation - CSE + GT BT”, which does not use ground-truth search locations?”
> >
> > We thank the reviewer for catching this and will update the improvement from 35.35 (Ours+GT Segmentation - CSE + GT BT) to 36.13 (Ours+GT Both - CSE + GT BT). With this correction, the improvement is larger than previously reported. Note that “CSE + GT BT” from Gadre et al., 2022 refers to CSR with ground truth (GT) object detection boxes (B) and navigation trajectories (T) that visit objects that have moved.

---

> > > ### Author Response · Authors · 2022-11-10
> > > **Response To Reviewer Vjug (3/3)**
> > >
> > > References:
> > >
> > > S. Y. Min, D. S. Chaplot, P. Ravikumar, Y. Bisk, and R. Salakhutdinov. FILM: following instructions in language with modular methods. In ICLR, 2022.
> > >
> > > V. Blukis, C. Paxton, D. Fox, A. Garg, and Y. Artzi. A persistent spatial semantic representation for high-level natural language instruction execution. In CoRL, 2021.
> > >
> > > D. S. Chaplot, M. Dalal, S. Gupta, J. Malik, and R. Salakhutdinov. SEAL: self-supervised embodied active learning using exploration and 3d consistency. In M. Ranzato, A. Beygelzimer, Y. N. Dauphin, P. Liang, and J. W. Vaughan, editors, NeurIPS, 2021.
> > >
> > > D. S. Chaplot, D. Gandhi, S. Gupta, A. Gupta, and R. Salakhutdinov. Learning to Explore using Active Neural SLAM. In ICLR, 2020.
> > >
> > > L. Weihs, M. Deitke, A. Kembhavi, and R. Mottaghi. Visual room rearrangement. In CVPR, 2021.
> > >
> > > S. Y. Gadre, K. Ehsani, S. Song, and R. Mottaghi. Continuous scene representations for embodied AI. In CVPR, 2022.

---

> ### Author Response · Authors · 2022-11-13
> **New Experiment For Noisy Pose Estimation**
>
> We have conducted an additional ablation where we simulate noisy pose estimation due to imperfect localization and mapping. We adopt the sensor noise model introduced by Sarch et al., 2022, which builds on earlier work from Chaplot et al., 2020, and constructs a Gaussian distribution based on real robot localization data collected from a real LocoBot robot (Murali et al., 2019), and applies this noise model in simulation. This model adds gaussian noise with $\sigma=0.005$ meters to positional observations, and adds gaussian noise with $\sigma=0.5$ degrees to yaw observations at each timestep.
>
> | Method            | %Fixed Strict (Val) | Success (Val) | %Fixed Strict (Test) | Success (Test) |
> |-------------------|---------------------|---------------|----------------------|----------------|
> | MaSS              | 17.47               | 6.30          | 16.56                | 4.70           |
> | MaSS + Noisy Pose | 19.84               | 6.80          | 17.33                | 4.90           |
>
> Shown above, our method is robust to this sensor noise model. On both the validation and test sets of the 2022 Rearrangement Challenge, sensor noise leads to a **slight improvement** in performance, which we attribute to noise producing a smoother map.
>
> We believe this new experiment resolves Reviewer Vjug’s concern of “whether the model can maintain effectiveness when facing … a noisy environment (noisy pose estimation).”
>
> References:
>
> G. Sarch, Z. Fang, A. W. Harley, P. Schydlo, M. J. Tarr, S. Gupta, K. Fragkiadaki. TIDEE: Tidying Up Novel Rooms using Visuo-Semantic Commonsense Priors. In ECCV, 2022.
>
> A. Murali, T. Chen, K. V. Alwala, D. Gandhi, L. Pinto, S. Gupta, A. Gupta. PyRobot: An Open-source Robotics Framework for Research and Benchmarking. ArXiv, 2019.
>
> D. S. Chaplot, D. Gandhi, S. Gupta, A. Gupta, and R. Salakhutdinov. Learning to Explore using Active Neural SLAM. In ICLR, 2020.

---

> ### Author Response · Authors · 2022-11-16
> **Requesting A Discussion, Reviewer Vjug**
>
> Please let us know whether our response has clarified any of your concerns. If not, please let us know what is not resolved and we can clarify any of the raised concerns. We look forward to having a discussion.

---

### Official Review · Reviewer_wg2G · 2022-10-24

**Confidence:** 4
**Correctness:** 4
**Technical Novelty And Significance:** 2
**Empirical Novelty And Significance:** 2
**Recommendation:** 8

**Clarity, Quality, Novelty And Reproducibility:**

The paper is clearly written. In terms of originality, most components in the paper were already proposed in previous work. However, the right combination of these different pieces to be applied to the rearrangement task can be considered as novel.

**Strength And Weaknesses:**

Strengths:
1. The improvement in performance brought from the method is significant. This is particularly important to note as the targeted task is hard.
2. The method is clearly presented in the paper, and the experiments are thorough.
3. Simplicity of the approach can be considered as a strength as it allows easier reproducibility.


Weaknesses:
1. Training policies to use and build semantic voxel maps of 3D scenes has already been done in prior work in Embodied AI. This is properly acknowledged by authors but a lack of novelty is still a major concern.
2. I am not convinced by the necessity to learn the semantic priors in the first phase (i.e. scene exploration from the policy). Wouldn’t a simple exploration (e.g. maximising coverage) of the full environment be the best thing to do as objects could be placed anywhere in the environment ?

**Summary Of The Paper:**

This paper targets the room rearrangement task, and proposes a method outperforming the current state-of-the-art end-to-end method. The approach introduced in this work is about building two voxel-based 3D semantic maps of the scene before and after (specified goal)  rearrangement to then rearrange objects with different location in the two maps. First, a trained policy navigates both scenes to build the maps with as many semantic objects as possible. Once the maps are built, disagreements are searched and corresponding objects are rearranged.  The method is pretty simple, but efficient as showcased in the experimental study. It can become a strong baseline for the task.

**Summary Of The Review:**

The major weakness of this work is the lack of novelty in the proposed method. However, as already mentioned, a proper combination of known methods to achieve a strong performance on a challenging task is still interesting for the community. Moreover, authors propose a detailed experiment section to evaluate the performance of their method, and provide promising future directions to build on top of their work by considering the upper bound performance when using ground-truth perception. I would thus tend to consider this work as marginally above the acceptance threshold, and am waiting for answers from the authors about my remarks.

---

> ### Author Response · Authors · 2022-11-10
> **Response To Reviewer wg2G**
>
> We thank Reviewer wg2G for their assessment of the paper.
>
> * “Training policies to use and build semantic voxel maps of 3D scenes has already been done in prior work in Embodied AI. This is properly acknowledged by authors but a lack of novelty is still a major concern.”
>
> Our novelty lies in the problem statement: rearranging objects to a goal state, when the goal state is only passively observed during an initial walkthrough phase. Prior works that build voxel-based semantic maps, including Min et al., 2022, Blukis et al. 2021, and Chaplot et al. 2021, assume the task goal is specified extrinsically. For example, in Min et al., 2022 and Blukis et al. 2021 the task goal is specified to the agent via a prompt. Similarly, in Chaplot et al. 2021 the task goal is specified via an object category. However, in our setting, the agent must act to observe the goal, and lacking proper exploration during the walkthrough phase, the agent may not observe the goal at all. Our unique problem statement requires our method to reason about scene changes. We are the first to consider the use of voxel-based semantic maps in a setting where the agent must explore and reason about scene changes to observe the task goal.
>
> Our goal is to find the simplest method capable of serving as a strong baseline for our unique problem statement. All reviewers agree the method’s simplicity is a strength. Our use of learned policies with voxel-based semantic maps stems from our desire for simplicity. Our method does not require more sophistication to perform well in practice.
>
> * “I am not convinced by the necessity to learn the semantic priors in the first phase (i.e. scene exploration from the policy). Wouldn’t a simple exploration (e.g. maximizing coverage) of the full environment be the best thing to do as objects could be placed anywhere in the environment ?”
>
> Previous work has shown that semantic priors help agents discover objects more quickly in scenes where objects are placed in natural locations (Min et al., 2022). For example, a knife is more likely to be found on the kitchen countertop than on the floor. The reviewer’s intuition is correct, that in Min et al., 2022, the target object is known, whereas in our setting the task goal is unknown during the first phase. Even for unconditional search, the goal of finding all objects in the scene is still well defined: the optimal policy would learn to search locations that commonly contain objects---the kitchen countertop---and ignore locations that do not commonly contain objects. Table 1 in our paper shows Semantic Search helps by +1.45 %Fixed Strict on the test set, and we are conducting an additional ablation that applies Semantic Search independently during either the first or second phases, and will add this to our final manuscript.
>
> References:
>
> S. Y. Min, D. S. Chaplot, P. Ravikumar, Y. Bisk, and R. Salakhutdinov. FILM: following instructions in language with modular methods. In ICLR, 2022.
>
> V. Blukis, C. Paxton, D. Fox, A. Garg, and Y. Artzi. A persistent spatial semantic representation for high-level natural language instruction execution. In CoRL, 2021.
>
> D. S. Chaplot, M. Dalal, S. Gupta, J. Malik, and R. Salakhutdinov. SEAL: self-supervised embodied active learning using exploration and 3d consistency. In M. Ranzato, A. Beygelzimer, Y. N. Dauphin, P. Liang, and J. W. Vaughan, editors, NeurIPS, 2021.

---

> > ### Comment · Reviewer_wg2G · 2022-11-14
> > **Response to authors**
> >
> > I thank the authors for addressing my concerns.
> >
> > - Assessing whether applying a well-known method to a new problem statement presents enough novelty is always a tricky question. However, I tend to think that proposing a simple method that outperforms previous baselines on a challenging task is a relevant contribution for the community and am happy with the answer from the authors.
> >
> > - I understand the intuition behind having a semantic search. However, the impact of using Semantic Search does not seem very strong to me according to the results in Table 1 (compared with the difference in performance between baselines and the method without Semantic Search). If authors have run different training runs (with different seeds), I would like them to present results in terms of metrics mean and standard deviation in order to assess whether the mentioned gain of +1.45% is significant. Finally, authors mention in their response that they “are conducting an additional ablation that applies Semantic Search independently during either the first or second phases, and will add this to our final manuscript.”. It would be great to have access to the results of this interesting ablation study before the end of the discussion period if possible.

---

> > > ### Author Response · Authors · 2022-11-15
> > > **New Experiment For Semantic Search**
> > >
> > > We thank Reviewer wg2G for discussing our rebuttal and for their interest in our Semantic Search ablation. We provide results below for this new ablation on the test set of the 2022 Rearrangement Challenge, where we apply our Semantic Search policy independently during either the first or second phases, but not both at once. The goal of this new ablation is to show that our Semantic Search leads to an improvement in performance in both cases. We also report a 68% confidence interval following the $\pm$ sign in each cell, computed across the 1000 tasks present in the test set. Results show that in all cases Semantic Search leads to an improvement, and terms of the *Success (Test)* metric, the improvement exceeds the 68% confidence interval.
> > >
> > > | Method                             | %Fixed Strict (Test) | Success (Test)  |
> > > |------------------------------------|----------------------|-----------------|
> > > | MaSS w/o Semantic Search             | 15.11 $\pm$ 0.84     | 3.60 $\pm$ 0.58 |
> > > | MaSS + Semantic Search Unshuffle   | 15.97 $\pm$ 0.87     | 4.40 $\pm$ 0.64 |
> > > | MaSS + Semantic Search Walkthrough | 16.16 $\pm$ 0.89     | 4.90 $\pm$ 0.68 |
> > >
> > > We believe this new experiment, reported as confidence intervals, addresses Reviewer wg2G’s aim in understanding “whether the mentioned gain of +1.45% is significant.”

---

> > > > ### Comment · Reviewer_wg2G · 2022-11-15
> > > > **Response to authors**
> > > >
> > > > I thank the authors for their additional work. A 68% confidence interval must be computed for all results in the main paper as it is very valuable (and such results must be reported in the main paper for all baselines if possible).
> > > >
> > > > Could you please explain in more details what "MaSS + Semantic Search Unshuffle" and "MaSS + Semantic Search Walkthrough" correspond to, and how they compare with the "Ours" line in Table 1 in the main paper? It indeed looks like "MaSS + Semantic Search Walkthrough" and "Ours" (Table 1 main paper) have comparable performance.

---

> > > > > ### Author Response · Authors · 2022-11-17
> > > > > **Further Ablation Of Semantic Search For Reviewer wg2G**
> > > > >
> > > > > We appreciate Reviewer wg2G’s suggestion, and highlight Appendix C: Performance Confidence Intervals in the paper. “Ours” attains `16.56 ± 0.89` %Fixed Strict (Test) and `4.70 ± 0.67` Success (Test). Our baselines Weihs et al., 2021 (VRR + Map) and Gadre et al., 2022 (CSR) only report single numbers, and we cite their numbers directly. We agree with the reviewer that Embodied AI papers should be reporting confidence intervals, but we do not feel equipped to reproduce these baselines to obtain them.
> > > > >
> > > > > * “Could you please explain in more detail what "MaSS + Semantic Search Unshuffle" and "MaSS + Semantic Search Walkthrough" correspond to, and how they compare with the "Ours" line in Table 1 in the main paper? It indeed looks like "MaSS + Semantic Search Walkthrough" and "Ours" (Table 1 main paper) have comparable performance.”
> > > > >
> > > > > “MaSS + Semantic Search Walkthrough” corresponds to an ablation where Semantic Search is used during the Walkthrough phase, and a baseline that samples navigation goals uniformly in free space is used during the other phase. “MaSS + Semantic Search Unshuffle” is defined similarly. These differ from “Ours”, which uses Semantic Search during both phases. “MaSS + Semantic Search Walkthrough” and “Ours” have similar performance likely because the Unshuffle phase is longer than the Walkthrough phase, reducing the need for efficient exploration.
> > > > >
> > > > > | Horizon                  | %Objects Found (Unshuffle Phase, Test)  |
> > > > > |--------------------------|-----------------------------------------|
> > > > > | By 100 Episode Timesteps | +6.23 $\pm$ 1.20                        |
> > > > > | By 250 Episode Timesteps | +2.62 $\pm$ 0.77                        |
> > > > > | By 500 Episode Timesteps | +1.67 $\pm$ 0.64                        |
> > > > >
> > > > > Summarized above, we conduct a new experiment in Appendix H: Effect Of Semantic Search On Found Objects, tracking the percent of objects found by the agent. When the budget is small (100 steps), Semantic Search leads to a gain of `+6.23 $\pm$ 1.20` over the baseline. When the budget is large (500 steps), that gain is reduced to `+1.67 $\pm$ 0.64`. Due to the longer episode length during the Unshuffle phase (500 steps), efficient exploration is less beneficial than it is during the Walkthrough phase.
> > > > >
> > > > > References:
> > > > >
> > > > > L. Weihs, M. Deitke, A. Kembhavi, and R. Mottaghi. Visual room rearrangement. In CVPR, 2021.
> > > > >
> > > > > S. Y. Gadre, K. Ehsani, S. Song, and R. Mottaghi. Continuous scene representations for embodied AI. In CVPR, 2022.

---

> > > > > > ### Comment · Reviewer_wg2G · 2022-11-18
> > > > > > **Response to authors**
> > > > > >
> > > > > > I thank the authors for the clarifications and additional experiments. This new study about the impact of Semantic Search on the percentage of found objects depending on the time budget provides interesting results and allows to better assess the importance of Semantic Search.
> > > > > >
> > > > > > After this fruitful discussion, I have updated my recommendation score to 8. Even though the technical novelty in this work is somewhat limited, I still think it provides a strong baseline for an important task, along with thorough experimental studies that are valuable.

---

### Official Review · Reviewer_jZAh · 2022-10-24

**Confidence:** 4
**Correctness:** 3
**Technical Novelty And Significance:** 3
**Empirical Novelty And Significance:** 3
**Recommendation:** 6

**Clarity, Quality, Novelty And Reproducibility:**

The work is sufficiently novel and well-written. I do not expect any issues with reproducibility.

**Strength And Weaknesses:**

Strengths:
- The improvement of the work over the previous state-of-the-art is very impressive.
- The idea is very simple and creative. It is also intuitively easy to visualize and understand failure cases since the internal representations are just voxel-based.
- The component ablation section is very interesting and well thought out.

Weaknesses:
- W1. The setup is a bit overengineered to the visual room rearrangement challenge, and doesn't seem all that robust to practical real-world rearrangement scenarios. Suppose there were small changes in the scenes that might occur when humans actually live in these rooms (e.g., background objects unexpectedly moved or the rest of the scene wasn't perfectly stable). It seems like this approach would have a pretty hard time.
- W2. Using voxel-based representations will scale poorly from a memory perspective to much larger scenes. For this task, it seems sufficiently workable, since all the scenes are 1-room scenes, but it seems like it might become very inefficient to scale beyond simple rooms.

**Summary Of The Paper:**

The authors propose an approach for the visual room rearrangement challenge that creates a voxelized map of the scene during the walkthrough and unshuffle phases, finds the differences between the maps, and trains an agent to rearrange the objects based on the differences. The approach improves substantially over the previous state-of-the-art for the 2-phase variant of the task.

**Summary Of The Review:**

In general, I'm positive about the paper. It is fairly creative and sufficiently novel. However, I get the sense that it's a bit over-engineered to solve the visual room rearrangement challenge. Regardless, I recommend an acceptance of the work.

---

> ### Author Response · Authors · 2022-11-10
> **Response To Reviewer jZAh**
>
> We thank Reviewer jZAh for their assessment of the paper.
>
> * “W1. The setup is a bit over engineered to the visual room rearrangement challenge, and doesn't seem all that robust to practical real-world rearrangement scenarios. Suppose there were small changes in the scenes that might occur when humans actually live in these rooms (e.g., background objects unexpectedly moved or the rest of the scene wasn't perfectly stable). It seems like this approach would have a pretty hard time.”
>
> We agree with the reviewer that the problem setting of the Visual Room Rearrangement challenge is not perfectly realistic, especially in its lack of humans alongside the agent. However, even with its simplifying assumptions Visual Room Rearrangement remains a very difficult problem, shown by the low performance of other Reinforcement Learning and Planning baselines used in prior work. This difficulty is acknowledged by the other two reviewers, where Reviewer wg2G states “The improvement in performance brought from the method is significant. This is particularly important to note as the targeted task is hard” and Reviewer Vjug states “performance improvements on the challenging task over a competing baseline are significant.” We believe in improving the realism of benchmarks as future work, but the current difficulty of the task suggests a need to develop more effective methods, even though some unrealistic assumptions are made.
>
> * “W2. Using voxel-based representations will scale poorly from a memory perspective to much larger scenes. For this task, it seems sufficiently workable, since all the scenes are 1-room scenes, but it seems like it might become very inefficient to scale beyond simple rooms.”
>
> Voxel-based scene representations are frequently used in embodied tasks, including the recent work of Min et al., 2022 and Blukis et al. 2021 which adopt voxel-based scene representations for single room tasks, and Chaplot et al. 2021, which adopts voxel-based maps in the Habitat simulator for the Gibson dataset. The Habitat Gibson dataset consists of 3D reconstructions of real-world homes with multiple rooms. Voxel maps “will scale poorly from a memory perspective” in the sense that as the spatial size of houses increases, the memory usage tends to increase quadratically. This is because floor-to-ceiling distance is typically constant. However, these prior works show voxel maps fit within existing device memory constraints even for large multi-room houses.
>
> References:
>
> S. Y. Min, D. S. Chaplot, P. Ravikumar, Y. Bisk, and R. Salakhutdinov. FILM: following instructions in language with modular methods. In ICLR, 2022.
>
> V. Blukis, C. Paxton, D. Fox, A. Garg, and Y. Artzi. A persistent spatial semantic representation for high-level natural language instruction execution. In CoRL, 2021.
>
> D. S. Chaplot, M. Dalal, S. Gupta, J. Malik, and R. Salakhutdinov. SEAL: self-supervised embodied active learning using exploration and 3d consistency. In M. Ranzato, A. Beygelzimer, Y. N. Dauphin, P. Liang, and J. W. Vaughan, editors, NeurIPS, 2021.

---

> > ### Comment · Reviewer_jZAh · 2022-11-16
> > **Response to authors**
> >
> > Thank you for your response.
> >
> > I'm generally positive about the paper, but still feel it's quite over-engineered to the visual room rearrangement challenge, and not all that practically useful in its current state. Nevertheless, I find the work sufficiently interesting, well-presented, and impressive with respect to performance. I still believe it should be accepted and will keep my rating.

---

> ### Author Response · Authors · 2022-11-16
> **Requesting A Discussion, Reviewer jZAh**
>
> Please let us know whether our response has clarified any of your concerns. If not, please let us know what is not resolved and we can clarify any of the raised concerns. We look forward to having a discussion.

---

### Author Response · Authors · 2022-11-10
**Thanks For The Valuable Feedback + Summary**

The authors thank all the reviewers for their valuable feedback. All reviewers unanimously agree on the value of simplicity of our approach, leading to large improvements. Reviewer jZAh states “the idea is very simple and creative. It is also intuitively easy to visualize and understand failure cases.” Reviewer Vjug states our method is “self-motivated and interesting.” Reviewer wg2G states the “simplicity of the approach can be considered as a strength as it allows easier reproducibility.”

We address specific questions below.

---

### Decision · Program_Chairs · 2023-01-20

**Decision:**

Accept: poster

**Justification For Why Not Higher Score:**

The 3D mapping component, which is claimed as a main contribution of the paper is almost identical to previous work (e.g., Blukis et al.).

**Justification For Why Not Lower Score:**

The paper includes valuable contributions so I vote for acceptance.

**Metareview: Summary, Strengths And Weaknesses:**

The paper proposes an approach to the Visual Room Rearrangement problem (Weihs et al.). First, a 3D semantic map of the scene is created. Using the map, the agent is able to identify the differences between the initial configuration and the goal configuration (that was observed during the walkthrough stage).

Strengths:
- The paper establishes a new state-of-the-art for the task of room rearrangement, and there is a large gap between the performance of this method and the previous approaches.
- The method is clearly described, and the experiments are well thought out.

Weaknesses:
- The mapping approach is almost identical to Blukis et al. The citation should be where the mapping approach is explained and not hidden in the related work section.
- The effectiveness of the approach depends on the perfect pose of the agent. (New results using noisy pose addressed this concern)

The reviewers found the contributions valuable for the community. The AC follows the recommendation of the reviewers assuming that proper citations will be provided for the Blukis et al. work.


**Note From Pc:**

if the above contains the word "oral" or "spotlight" please see: "oral" presentation means -> notable-top-5% and "spotlight" means -> notable-top-25%. As stated in our emails, we are disassociating presentation type from AC recommendations